# Preparation of Biochar Composite Microspheres and Their Ability for Removal with Oil Agents in Dyed Wastewater

**DOI:** 10.3390/ma16186155

**Published:** 2023-09-11

**Authors:** Lu Shen, Rushi Jin, Wanming Chen, Dongming Qi, Shimin Zhai

**Affiliations:** 1Key Laboratory of Advanced Textile Materials and Manufacturing Technology, Ministry of Education, Zhejiang Sci-Tech University, Hangzhou 310018, China; zhelintjw@163.com (L.S.); jinrushi@126.com (R.J.); dongmingqi@zstu.edu.cn (D.Q.); 2Zhejiang Haoyu Technology Co., Ltd., Shaoxing 312000, China; wanming@zhejianghaoyu.com; 3Key Laboratory of Green Cleaning Technology & Detergent of Zhejiang Province, Lishui 323000, China

**Keywords:** biomass resource utilization, biochar microspheres, structural modification, oily dye wastewater treatment

## Abstract

Oil agents produced from the degreasing treatment of synthetic fibers are typical pollutants in wastewater from printing and dyeing, which may cause large-scale environmental pollution without proper treatment. Purifying oily dye wastewater (DTY) at a low cost is a key problem at present. In this study, biochar microspheres with oil removal ability were prepared and derived from waste bamboo chips using the hydrothermal method. The structure of the biochar microsphere was regulated by activation and modification processes. Biochar microspheres were characterized, and their adsorption behaviors for oily dye wastewater were explored. The results show that the adsorption efficiency of biochar microspheres for oily dye wastewater (DTY) was improved significantly after secondary pyrolysis and the lauric acid grafting reaction. The maximum COD removal quantity of biochar microspheres for DTY was 889 mg/g with a removal rate of 86.06% in 30 min. In addition, the kinetics showed that chemisorption was the main adsorption manner. Considering the low cost of raw materials, the application of biochar microspheres could decrease the cost of oily wastewater treatment and avoid environmental pollution.

## 1. Introduction

With the continuous development of social economy and industry, freshwater resources are becoming increasingly scarce with extremely low per capita water availability, uneven spatial distribution, and serious water pollution, which have become the main factors restricting sustainable development [1,2]. With an increasing awareness of environmental protection, national requirements for the discharge of wastewater from the textile printing and dyeing industry are also becoming higher. How to efficiently and cost-effectively treat wastewater from printing and dyeing is a difficult problem faced by current textile printing and dyeing enterprises, among which oil dispersants are one of the typical pollutants in wastewater from printing and dyeing [3]. The highest allowable discharge concentration of oily wastewater in China is only 1 mg/L, and due to its complex composition, there is a difficulty in demulsification with the and easy introduction of new pollutants by demulsification, which are difficult and costly to treat [4]. Biochar materials, due to their large surface area, high porosity, rich surface functional groups, and strong adsorption and removal ability for organic pollutants, have been widely used in the field of wastewater treatment [5]. In addition, biochar materials have a wide range of raw material sources and low cost, and their application in the treatment of wastewater from printing and dyeing can effectively reduce the cost of wastewater treatment and realize the resource utilization of biomass [6].

Synthetic fibers are hydrophobic polymer materials that can be used in oil dispersants as smoothing agents, antistatic agents, emulsifiers, etc., in spinning and textile processing [7]. Before the dyeing process, the oil dispersant needs to be removed in order to carry out normal dyeing, which results in a large number of oil dispersants in the wastewater from printing and dyeing. Because the oil dispersants in synthetic fibers are rarely composed of a single chemical structure material but rather require various substrates (surfactants, mineral oils, advanced alcohols, fatty acid esters, etc.) to be combined according to their respective characteristics and purposes, this results in a high COD value, strong biological toxicity, difficulty in demulsification, the difficulty and cost of treatment, and easily causes large-scale environmental pollution [8]. However, common degreasing methods have found it hard to break emulsion with the addition of surfactants. A lot of research has been conducted to improve the efficiency of oily wastewater treatment. Ao et al. [9] prepared amine-functionalized cotton to treat oily wastewater, and the de-oiling rate reached 98.5%. Gholamifard et al. [10] used natural zeolite and calcined bentonite to treat oily wastewater with a maximum capacity of 3.06 mg/g and 5.37 mg/g, respectively. However, high cost and low efficiency were still the main problems during the oily wastewater treatment process [11,12]. Therefore, preparing a cheap oil removal adsorbent is a pressing issue. Therefore, biochar has the potential to be used in the treatment of oily wastewater as an inexpensive adsorbent. For example, Sarup et al. [13] reported that a superhydrophobic-oleophilic polyurethane sponge adsorbed oil from wastewater, in which the sponge was refurbished by biochar deriving from textile sludge. 

Herein, biochar microspheres deriving from waste bamboo were prepared using the hydrothermal method. After activation and modification, the magnetic lipophilic biochar microspheres (Fe_3_O_4_@L-ABM_500_) with a high degree of graphitization were obtained. The (Fe_3_O_4_@L-ABM_500_) displayed excellent removal ability for DTY oil in wastewater from dyeing and finishing. The application of biochar microspheres with high removal efficiency and low preparation cost could promote the large-scale utilization of waste biomass and decrease the treatment cost of oily wastewater. A low treatment cost would be beneficial to wastewater treatment plants. 

## 2. Experimental Section

### 2.1. Materials and Chemicals

Phloroglucinol (C_6_H_6_O_3_, AR grade), the iron sulfate heptahydrate (FeSO_4_•7H_2_O, AR grade) and ferric chloride hexahydrate (FeCl_3_•6H_2_O, AR grade), were purchased from National Drug Group Chemical Reagent Co., Ltd. (Shanghai, China). Lauric acid (C_12_H_24_O_2_, AR grade) was purchased from Tianjin Kemiou Chemical Reagent Co., Ltd. (Tianjin, China). Ammonium hydroxide (NH_3_•H_2_O, 28%) was purchased from Macklin Biochemical Co., Ltd. (Shanghai, China). Polyester oil (DTY, Commercial grade) was purchased from Huiya Environmental Protection Technology Co., Ltd. (Hangzhou, China).

Simulated oily dyed wastewater: 0.1 mL oil agent, 0.05 g disperse red dye, and 0.05 g sodium dodecyl sulfate were added into 100 mL of deionized water and then stirred for 30 min. The simulated oily wastewater from printing and dyeing with a COD concentration of 3150 mg/L was obtained and used in this study. 

### 2.2. Preparation of Biochar Composite Microspheres

Firstly, 3.2 g of bamboo flakes and 0.8 g of phloroglucinol were added into 120 mL of distilled water and stirred at 60 °C for 2 h. Then, the mixture was put into a hydrothermal reactor and pyrolyzed at 200 °C for 24 h with a heating rate of 1.5 °C/min in the muffle furnace. The mixture was ultrasonic-treated for 30 min at 500 W to remove carbonized bamboo pieces and large particle aggregates through a 240-mesh screen. After centrifugation at 4000 r/min for 10 min, the biochar microsphere (BM) was obtained. 

In total, 0.2 g of biochar microsphere (BM) was added into the KOH solution with a mass fraction of 10% and stirred for 10 min at 500 rpm. After being dried at 60 °C, the BM was pyrolyzed at different temperatures (300 °C, 500 °C and 700 °C) in the tube furnace with a N_2_ atmosphere for two hours, respectively. The activated biochar microspheres (ABM_300_, ABM_500_, ABM_700_) were obtained after washing with distilled water three times. Then, the prepared ABM_300_, ABM_500_ and ABM700 (0.2 g) were mixed with lauric acid (0.1 g) and pyrolyzed with the above conditions. Hydrophobic-modified biochars (L-ABM_300_, L-ABM_500_, L-ABM_700_) were obtained. 

Finally, 0.2 g L-ABM_300_, L-ABM_500,_ and L-ABM_700_ were dispersed into 80 mL of the iron–ion solution (50 °C) with 0.24 g of FeSO_4_•7H_2_O and 0.19 g of FeCl_3_•6H_2_O, respectively. The pH of the iron–ion solution was adjusted to 8.5 with ammonium hydroxide and cured for 5 min. Magnetic biochar microspheres (Fe_3_O_4_@L-ABM_300_, Fe_3_O_4_@L-ABM_500_, Fe_3_O_4_@L-ABM_700_) were prepared after washing with distilled water 3 times. 

### 2.3. Adsorption Tests

In total, 0.03 g of Fe_3_O_4_@L-ABM_300_, Fe_3_O_4_@L-ABM_500_, and Fe_3_O_4_@L-ABM_700_ were added into the simulated oily wastewater from dying (10 mL) and stirred for 30 min, respectively. Then, the mixture was centrifuged at 4000 rpm for 10 min to remove the biochar microsphere. The COD concentration of the residual solution was tested, and the removal rate and adsorption quantity were calculated. 

Then, 0.01 g, 0.02 g, 0.03 g, 0.04 g, and 0.05 g of Fe_3_O_4_@L-ABM_500_ were added into 10 mL of the oily wastewater from dying and stirred for 30 min. Based on the above results, the effects of different times (10 min, 20 min, 30 min, 40 min, and 50 min) and initial pH values (5–9.5) on the adsorption capability of biochar microspheres under the same condition were also investigated. The COD concentration of the residual solution was tested as aforementioned. 

### 2.4. Adsorption Kinetics

The adsorption kinetics were fitted by the Pseudo-first-order and Pseudo-second-order models as follows [14,15]:

Pseudo-first-order model:(1)$$\ln\left(Q_{e}-Q_{t}\right)=lnQ_{e}-K_{1}t$$

Pseudo-second-order model:(2)$$\frac{t}{Q_{t}}=\frac{1}{K_{2}Q_{e}^{2}}+\frac{t}{Q_{e}}$$

In Equations (1) and (2), Q_t_ (mg/g) is the adsorbance at t (min), Q_e_ (mg/g) is the saturated adsorption quantity, and K_1_ (min^−1^) and K_2_ (g/(mg·min)) are the equation constants, respectively. t (min) is the absorbance time.

### 2.5. Analytical Methods

Scanning electron microscopies (SEM) of biochar microspheres were performed using the SEM (SU8010, Hitachi, Japan). Fourier transform infrared (FTIR) spectra were characterized by an FTIR apparatus (Nicolet iS50, Thermo, Waltham, MA, USA). XPS was performed on a photoelectron spectrometer (ESCALAB 250XI, Thermo, Waltham, MA, USA) with a monochromated Al- K_α_ source at a residual gas pressure of less than 10^−8^ Pa. All the binding energies were referenced to the C 1 s peak at 284.6 eV of the surface’s adventitious carbon. The X-ray diffractions (XRD) of the bio-char materials were conducted using an X-ray diffractometer (D8, Brook AXS Co., Ltd., Karlsruhe, Germany) in the range of 15°–80° at the rate of 0.1°/min). The pore property of Fe_3_O_4_@L-ABM_500_ was characterized by a surface area analyzer (Tristar Ⅱ 3020, Quantachrome Instrument Co., Ltd., Boynton Beach, FL, USA) using the nitrogen gas adsorption isotherms. The residual COD concentration was measured using the potassium dichromate method (HJ 828-2017). All other parameters were measured following standard methods [16,17].

## 3. Results and Discussion

### 3.1. Characterizations

The morphologies of BM, ABM_500_ and Fe_3_O_4_@L-ABM_500_ were characterized by SEM images (Figure 1). 

As shown in Figure 1a, the spherical biochar (BM) with the size of 0.5–2 μm was successfully prepared using the hydrothermal carbonization method from bamboo slices, which had a smooth surface. The biochar microsphere (ABM_500_), activated by KOH, is shown in Figure 1b. It can be seen that a large number of structural defects were generated on the surface of the biochar microsphere. Structural defects will increase the specific surface area of the microsphere and improve the contact area of the oil agent [18,19]. A high surface area can promote the grafting reaction with lauric acid and improve its adsorption capacity. An SEM image of Fe_3_O_4_@L-ABM_500_ is displayed in Figure 1c. More defects occurred, and many nanoparticles were loaded on the microspheres. The nanoparticles around the microspheres could be attributed to Fe_3_O_4_ components.

Raman analyses for Fe_3_O_4_@L-ABM_300_, Fe_3_O_4_@L-ABM_500,_ and Fe_3_O_4_@L-ABM_700_ were conducted to explore the influences of pyrolysis temperature on the biochar microspheres (Figure 2). In Figure 2, Fe_3_O_4_@L-ABM_300_, Fe_3_O_4_@L-ABM_500,_ and Fe_3_O_4_@L-ABM_700_ all showed characteristic peaks at around 1350 cm^−1^ and 1580 cm^−1^, corresponding to the D band and G band, respectively. Herein, the D band represents disordered graphite carbons, and the G band reflects the sp^2^ hybrid carbon–carbon bond resonance of graphite crystalline. The higher intensity ratio of characteristic peaks (I_D_/I_G_) indicates the greater crystal defects and disordered carbon components that occurred in the carbon material. In Figure 2a–c, the I_D_/I_G_ values of Fe_3_O_4_@L-ABM_300_, Fe_3_O_4_@L-ABM_500,_ and Fe_3_O_4_@L-ABM_700_ were 1.31, 0.88, and 0.85, respectively. The significant decrease in I_D_/I_G_ values indicated that the higher pyrolysis temperature improved the graphitization degree of biochar microspheres. For the higher degree of graphitization, the biochar microspheres might have had a stronger affinity for oil agents [20,21].

To explore the crystal structures and functional groups of biochar microspheres when prepared at different conditions, X-ray diffractions, and FTIR spectra analyses were conducted, as shown in Figure 3. In Figure 3a, the BM, Fe_3_O_4_@ABM_500,_ and Fe_3_O_4_@L-ABM_500_ all displayed obvious gentle peaks at 23.1°, which could be ascribed to the amorphous carbon in biochar microspheres. After pyrolysis at 500 °C, the peak intensity (23.1°) of Fe_3_O_4_@ABM_500_ and Fe_3_O_4_@L-ABM_500_ was weaker than that of BM, indicating that the pyrolysis process improved the graphitization degree of biochar. This result is consistent with the above Raman analysis. Compared with BM, Fe_3_O_4_@ABM_500_ and Fe_3_O_4_@L-ABM_500_ appeared with new characteristic peaks at 30.1°, 35.5°, 43.1°,53.4°,57.0° and 62.6°, which could be attributed to the (220), (311), (400), (422), (511) and (440) planes of Fe_3_O_4_ (, respectively [22]. These indicate that Fe_3_O_4_ was successfully loaded on the activated biochar microspheres. The biochar composite materials with Fe_3_O_4_ in situ loaded could be separated rapidly from the clean water in the presence of a magnetic field, with the potential to improve adsorption efficiency.

The FTIR spectra of BM, ABM_500_, L-ABM_300_, L-ABM_500,_ and L-ABM_700_ are listed in Figure 3b. The peaks at 3411 cm^−1^ could be ascribed to the stretching vibration (υ_O-H_) of the hydroxyl group (O-H), and the two peaks occurred in a range of 2900 cm^−1^–2670 cm^−1^ corresponding to the stretching vibration (υ_C-H_) of the methyl group (-CH_3_). The peaks at 1628 cm^−1^ and 1383 cm^−1^ indicate the presence of the stretching vibration (υ_C=C_) of C=C and the in-plane bending vibration (δ_C-H_) of C-H, respectively [23]. The biochar microspheres showed similar functional groups before and after modification. This is because the carboxylic groups and alkane structures involved in lauric acid also exist in the biochar microsphere. However, the intensity of the characteristic peaks was different. Compared with BM, the intensity of υ_C-H_ and δ_C-H_ in ABM_500_ decreased significantly, indicating a reduction in -CH_3_ and -CH_2_ contents after pyrolysis. These also indicated that the pyrolysis process improved the graphitization degree of the biochar microsphere. In addition, the intensity of υ_C-H_ and δ_C-H_ in L-ABM_500_ was stronger than that of ABM_500_, which could be attributed to the alkane structures introduced from lauric acid. This could be evidence that the grafting reaction between lauric acid and biochar microsphere was successful.

To obtain the chemical composition of as-prepared biochar microspheres, XPS spectra were performed, as shown in Figure 4. In Figure 4a, all three biochar microspheres (BM, Fe_3_O_4_@ABM_500,_ and Fe_3_O_4_@L-ABM_500_) had characteristic peaks at 286.3 eV and 531.9 eV, corresponding to the binding energies of C1s and O1s, respectively [24]. This means that the biochar microspheres had oxygen-containing functional groups expecting the graphite structure. Considering the FTIR spectra analysis (Figure 3b), the oxygen-containing functional groups involved the hydroxyl group (O-H), a carboxyl group (-COOH), and so on.

Compared with BM, Fe_3_O_4_@ABM_500_ and Fe_3_O_4_@L-ABM_500_ showed a new characteristic peak at 712.3 eV, which corresponded to the binding energy of Fe 2p in loaded Fe_3_O_4_. In addition, high-resolution spectra of Fe 2p in Fe_3_O_4_@ABM500 and Fe_3_O_4_@L-ABM_500_ were also conducted (Figure 4b). In Figure 4b, the characteristic peaks appeared at 710.2 eV and 723.5 eV, corresponding to the binding energies of Fe 2p_3/2_ and Fe 2p_1/2_ in Fe_3_O_4_, respectively [25]. This indicates that Fe_3_O_4_ was successfully loaded onto the biochar microspheres. This result is consistent with the XRD analysis results.

### 3.2. Adsorption Process Analysis

The influences of pyrolysis time, biochar dosage, adsorption time, and the initial pH on the adsorption processes are shown in Figure 5. In Figure 5a, it can be seen that the maximal COD removal rate of Fe_3_O_4_@L-ABM_500_ for oily wastewater reached 85.5% with an adsorption quantity of 887.3 mg/g. The Fe_3_O_4_@L-ABM_500_ pyrolyzed at 500 °C with a higher COD removal rate than the Fe_3_O_4_@L-ABM_300_ and Fe_3_O_4_@L-ABM_700_, which pyrolyzed at 300 °C and 700 °C, respectively. This phenomenon could be ascribed to the graphitization degree and functional group content of biochar microspheres. This is because Fe_3_O_4_@L-ABM_300_ was pyrolyzed at a relatively low temperature (300 °C). Fe_3_O_4_@L-ABM_300_ has a low graphitization degree, resulting in poor lipophilicity and weak adsorption ability of oily wastewater. With the increase in pyrolysis temperature, the graphitization degree of biochar microspheres improved, which could contribute to the adsorption ability of oily wastewater. However, the higher the pyrolysis temperature, the fewer functional groups there were reserved in the biochar microspheres. The high pyrolysis temperature (700 °C) of Fe_3_O_4_@L-ABM_700_ was adverse to the grafting reaction of lauric acid onto biochar microspheres. Hence, Fe_3_O_4_@L-ABM_700_ had a lower adsorption ability than Fe_3_O_4_@L-ABM_700_.

The adsorption ability of Fe_3_O_4_@L-ABM_500_ for oily wastewater with different dosages is displayed in Figure 5b. It can be seen that the COD removal rate increased with the increase in biochar dosage. When the dosage of Fe_3_O_4_@L-ABM_500_ was 0.03 g, the COD removal rate reached 86.06% with an adsorption quantity of 1037.2 mg/g. However, the COD removal rate decreased with the dosage, which continuously increased from 0.03 g to 0.05 g. This could be ascribed to the ash and unreacted lauric acid on biochar microspheres dissolved into the solution, resulting in an increase in COD concentrations.

In addition, the effects of adsorption time and initial pH on the adsorption abilities are also explored in Figure 5c,d. In Figure 5c, the COD removal rate increased with the increase in adsorption time and reached the adsorption equilibrium (84.6%) gradually at 30 min with the adsorption quantity of 889 mg/g. These results indicate that 30 min is a suitable adsorption time, and the Fe_3_O_4_@L-ABM_500_ had a great adsorption efficiency for oily wastewater. In Figure 5d, it can be seen that the initial pH affected the adsorption ability significantly. When the initial pH was 5.8, Fe_3_O_4_@L-ABM_500_ showed the highest adsorption capacity for oily wastewater, in which the COD removal rate was 84.5% (887 mg/g). However, the COD removal rate decreased significantly with the increase in the pH value. When the pH was 9.5, the COD removal rate decreased to 73%. Because the carboxyl groups (-COOH) were converted into -COO^-^ under alkaline conditions, this promoted its water solubility. Hence, the affinity of Fe_3_O_4_@L-ABM_500_ for oil decreased.

### 3.3. Adsorption Kinetics

The adsorption kinetic behaviors of Fe_3_O_4_@L-ABM_500_ for oily wastewater are explored in Figure 6, and the relative parameters are displayed in Table 1. It can be seen that the linear correlation coefficients (R^2^) of the Pseudo-first-order equation and Pseudo-second-order equation were 0.79 and 0.99, respectively. The great linear relationship between the t/Q_t_ value and adsorption time (t) indicated that the adsorption process was more suitable for the Pseudosecond-order equation.

Moreover, the equilibrium absorption quantity (Q_e_) of the Pseudo-second-order equation in Table 1 was also close to the experimental value. This is because the Pseudo-second-order equation assumed that the adsorption rate was linearly related to the concentration of the two reactants, reflecting the chemisorption process [26,27]. Hence, the chemisorption deriving from the introduced alkyl chains was the main adsorption manner of Fe_3_O_4_@L-ABM_500_ for oily wastewater. In addition, the pore properties of Fe_3_O_4_@L-ABM_500_ were also tested. The BET surface area, pore volume, and pore size of Fe_3_O_4_@L-ABM_500_ were 3.280 m^2^/g, 0.003 cm^3^/g, and 4.139 nm, respectively (See in the Appendix A).

Based on the above discussion and analysis, the possible adsorption mechanism of an oil agent onto the biochar microsphere is proposed. The adsorption process can be divided into three steps, including surface adsorption, inner diffusion, and adsorption binding onto the biochar. Firstly, the oil agent can be transferred easily from the solution to the outer surface of biochar due to the alkyl chain on the surface of biochar Then, the oil agent can further diffuse into the micropore of the biochar microsphere from the outer surface to the inner surface. Thus, the oil agent was absorbed into biochar microspheres efficiently. In this process, the introduced alkyl chains from lauric acid, pore structures, and a high degree of graphitization improved its adsorption efficiency [28].

## 4. Conclusions

In this study, the biochar microspheres were prepared from waste bamboo by a hydrothermal reaction, KOH activation, and lauric acid grafting modification. The prepared biochar microsphere (Fe_3_O_4_@L-ABM_500_) had high adsorption efficiency for oily wastewater, and the maximum COD removal rate reached 86.06%. Moreover, biochar microspheres (Fe_3_O_4_@L-ABM_500_) could be quickly separated from the solution under a magnetic field. Structural etching, graphitization degree, and lauric acid grafting played an important role in the adsorption process. Based on adsorption kinetics analysis, chemisorption was the main manner of adsorption. The high removal efficiency (COD removal rate of 86.06%) and low preparation costs of biochar microsphere (Fe_3_O_4_@L-ABM_500_) could promote the large-scale utilization of waste biomass and decrease the treatment cost of oily wastewater. However, the biochar microspheres after hydrothermal and modification treatment cannot remove oil completely. How to improve the absorption capacity of biochar microspheres in oil still needs further research.

## Figures and Tables

**Figure 1 materials-16-06155-f001:**
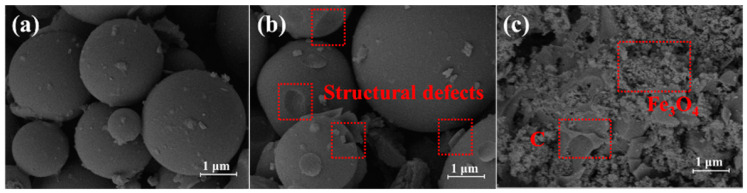
SEM images of BM (**a**), ABM_500_ (**b**) and Fe_3_O_4_@L-ABM_500_ (**c**).

**Figure 2 materials-16-06155-f002:**
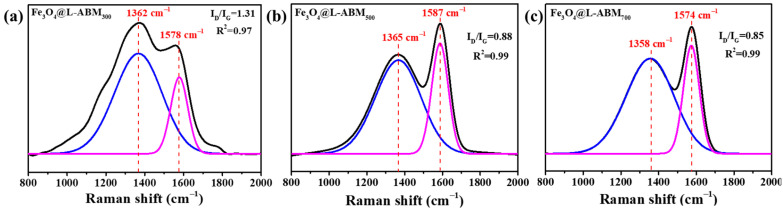
Raman analyses for Fe_3_O_4_@L-ABM_300_ (**a**), Fe_3_O_4_@L-ABM_500_ (**b**), Fe_3_O_4_@L-ABM_700_ (**c**).

**Figure 3 materials-16-06155-f003:**
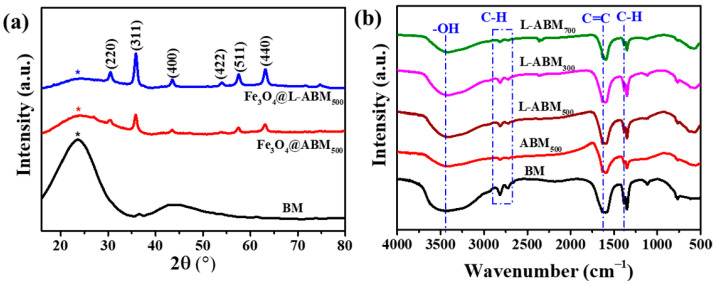
XRD patterns (**a**) and FTIR spectra (**b**) of as-prepared biochar microspheres.

**Figure 4 materials-16-06155-f004:**
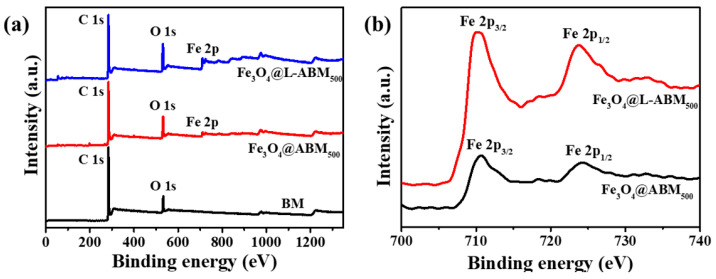
XPS survey spectra of BM, Fe_3_O_4_@ABM_500_ and Fe_3_O_4_@L-ABM_500_ (**a**); High-resolution spectra of Fe 2p in Fe_3_O_4_@ABM_500_ and Fe_3_O_4_@L-ABM_500_ (**b**).

**Figure 5 materials-16-06155-f005:**
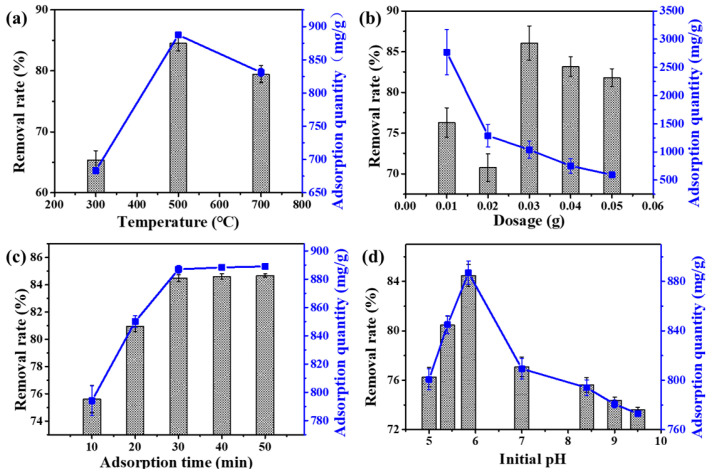
Effects of pyrolysis time (**a**), biochar dosage (**b**), adsorption time (**c**) and initial pH (**d**) on the adsorption abilities of biochar microspheres for oily wastewater.

**Figure 6 materials-16-06155-f006:**
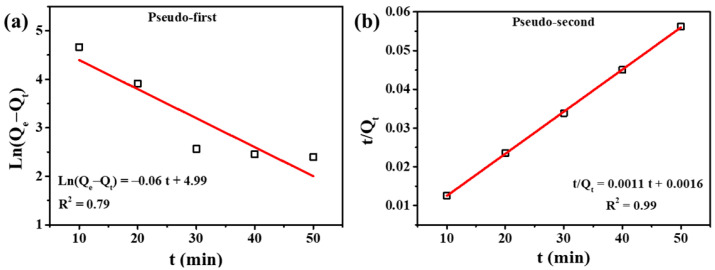
(**a**,**b**) Adsorption kinetic model-fitting curves of Fe_3_O_4_@L-ABM_500_ for oily wastewater.

**Table 1 materials-16-06155-t001:** Adsorption kinetics parameters of Pseudo-first- and Pseudo-second-order models.

Model	Fitting Parameters	Values
Pseudo-first-order equation	K_1_ (min^−1^)	0.06
	Q_e_ (mg/g)	146.94
	R^2^	0.79
Pseudo-second-order equation	K_2_ (g/(mg·min))	7.56 × 10^−4^
	Q_e_ (mg/g)	909.11
	R^2^	0.99

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
