# Peer review of "Preparation of Biochar Composite Microspheres and Their Ability for Removal with Oil Agents in Dyed Wastewater"

_materials, 2023, doi:10.3390/ma16186155_

Round 1

Reviewer 1 Report

Abstract:

Could you please provide the definition for the abbreviation "DTY" the first time it was introduced in the abstract (line 13)? Additionally, it would be helpful if the explanation could clearly outline how this abbreviation is derived.

In the abstract, the statement "can decrease the costs of oily wastewater treatment" raises a question: how can this cost reduction be achieved if the paper does not delve into its study or investigation?

Introduction:
While you highlight the challenges of wastewater treatment and the significance of biochar materials, explicitly state how the synthesis of biochar composite microspheres addresses a specific gap in the existing literature. What unique contribution does this study make?

Please provide a detailed literature review of similar research works done.

Clearly state what makes your study innovative. Is it the waste bamboo-derived source? Is it the specific modification technique? Highlight the novel aspects that differentiate your study from previous work.

Explicitly outline the objectives of the study. What are you aiming to achieve?

Explain why a high degree of graphitization is important for the magnetic lipophilic biochar microspheres. How does this attribute enhance their oil removal efficiency? Provide a brief rationale.

Acknowledge any limitations of the hydrothermal method and modification process.

How could industries or wastewater treatment plants benefit from this technology? Explain the potential positive impacts.

Methodology:
Briefly explain how COD of the samples were measured.

Conclusion:
Instead of stating a general "high adsorption efficiency," provide the actual percentage or value achieved.
Emphasize the significance of the biochar microspheres' magnetic separation capability. Explain how this characteristic enhances their practical applicability in real-world scenarios.
Suggest possible future research avenues building upon your work.
End your conclusion with a final statement that reiterates the significance of your study and its potential contributions to the field of wastewater treatment.
Instead of just mentioning "chemisorption," provide a brief explanation of what this term entails in this study.

Conduct a thorough proofread to eliminate grammatical errors and ensure consistency in formatting and style.

Author Response

Dear editor,

We would like to express our great appreciation to you and reviewers for the comments on our manuscript (Manuscript ID: materials-2580152, Title: Preparation of biochar composite microspheres and their removal ability for oil agents in dyeing wastewater). We have modified the manuscript accordingly and made revision which marked in yellow in the paper. Attached please find the revised version, which we would like to resubmit for your kind consideration as an article in Materials.

To facilitate your review, the detailed corrections are listed in the response document point by point. Both of marked manuscript and unmarked manuscript are supplied in the manuscript document. Looking forward to hearing from you.

With best regards,

Respectfully yours,

Shimin Zhai

Key Laboratory of Advanced Textile Materials and Manufacturing Technology, Ministry of Education

Zhejiang Sci-Tech University

928 the second street, Hangzhou 310018, China

Response to Reviewer 1 Comments 

Abstract:

Point 1: Could you please provide the definition for the abbreviation "DTY" the first time it was introduced in the abstract (line 13)? Additionally, it would be helpful if the explanation could clearly outline how this abbreviation is derived.

Response 1: Thank you for your advice. We have provided the definition for the abbreviation "DTY" the first time it was introduced in the abstract (line 13) in the revised manuscript. Besides, the “DTY” was the commercial abbreviation for this oil used in the processing of polyester.

Point 2: In the abstract, the statement "can decrease the costs of oily wastewater treatment" raises a question: how can this cost reduction be achieved if the paper does not delve into its study or investigation?

Response 2: Thank you for your question. In this study, the biochar microspheres were prepared from waste bamboo chips belonging to the agricultural wastes. Considering the low-cost of raw materials, the application of biochar microspheres may decrease the costs of oily wastewater treatment and avoid environmental pollution. We have modified in the revised manuscript as follows:

(Line 21): “Considering the low-cost of raw materials, the application of biochar microspheres may decrease the costs of oily wastewater treatment and avoid environmental pollution”

Introduction:

Point 3: While you highlight the challenges of wastewater treatment and the significance of biochar materials, explicitly state how the synthesis of biochar composite microspheres addresses a specific gap in the existing literature. What unique contribution does this study make?

Response 3: Thank you for your advice. We have stated the contribution of biochar composite microspheres in the revised manuscript as follows:

(Line 66-Line 68): “The application of biochar microspheres with high removal efficiency and low prepar-ing cost may promote the large-scale utilization of waste biomass, and decrease the treatment cost of oily wastewater.”

Point 4: Please provide a detailed literature review of similar research works done.

Response 4: Thank you for your advice. We have added detailed literature review of similar research works in the revised manuscript as follows:

(Line 57-Line 61): ‘Thereinto, biochar is potential to be used in the treatment of oily wastewater as an in-expensive adsorbent. For example, Sarup et al [] reported a superhydrophobic-oleophilic polyurethane sponge to adsorb oil from wastewater, in which the sponge was refur-bished by biochar deriving from textile sludge.’

Point 5: Clearly state what makes your study innovative. Is it the waste bamboo-derived source? Is it the specific modification technique? Highlight the novel aspects that differentiate your study from previous work.

Response 5: Thank you for your advice. In this study, the cheap oil removal adsorbent of biochar was prepared by lauric acid grafting reaction, and its application may effectively reduce the cost of oily wastewater treatment and promote the resource utilization of biomass. We highlighted the novel aspects in the revised manuscript as follows:

(Line 66-Line 68): “The application of biochar microspheres with high removal efficiency and low prepar-ing cost may promote the large-scale utilization of waste biomass, and decrease the treatment cost of oily wastewater.”

Point 6: Explicitly outline the objectives of the study. What are you aiming to achieve?

Response 6: Thank you for your advice. We emphasized the objectives of the study in the modified manuscript as follows:

(Line 56-Line 57): “Therefore, how to prepare a cheap oil removal adsorbent is a meaningful thing.”

Point 7: Explain why a high degree of graphitization is important for the magnetic lipophilic biochar microspheres. How does this attribute enhance their oil removal efficiency? Provide a brief rationale.

Response 7: Thank you for your question. The higher degree of graphitization, the less hydrophilic functional groups (-OH, -COOH et. al) in biochar microspheres.

Point 8: Acknowledge any limitations of the hydrothermal method and modification process.

Response 8: Thank you for your advice. We have added the limitations of the hydrothermal method and modification process in the revised manuscript as follows:

(Line 288-Line 291): “However, the biochar microspheres after hydrothermal and modification treatment cannot remove oil completely. How to improve the absorption capacity of biochar mi-crospheres for oil still needs further research.”

Point 9: How could industries or wastewater treatment plants benefit from this technology? Explain the potential positive impacts.

Response 9: Thank you for your advice. The benefits from this technology were displayed in the revised manuscript as follows:

(Line 66-Line 69): “The application of biochar microspheres with high removal efficiency and low prepar-ing cost may promote the large-scale utilization of waste biomass, and decrease the treatment cost of oily wastewater. The low treatment cost was beneficial to the wastewater treatment plants.”

Methodology:

Point 10: Briefly explain how COD of the samples were measured.

Response 10: Thank you for your advice. The COD concentrations of samples were tested by potassium dichromate method (HJ 828-2017) in the manuscript. We have added the information in the revised manuscript as follows:

(Line 130-Line 131): “The residual COD concentration was measured using potassium dichromate method (HJ 828-2017).”

Conclusion:

Point 11: Instead of stating a general "high adsorption efficiency," provide the actual percentage or value achieved.

Response 11: Thank you for your advice. We have modified in the manuscript as follows:

(Line 287): “The high removal efficiency (COD removal rate of 86.06%) and low preparing cost of biochar microsphere (Fe3O4@L-ABM500) may promote the large-scale utilization of waste biomass.”

Point 12: Emphasize the significance of the biochar microspheres' magnetic separation capability. Explain how this characteristic enhances their practical applicability in real-world scenarios.

Response 12: Thank you for your question. The precipitation method is the common separation method of used biochar. Herein, the magnetic biochar microspheres can be separated from solution under magnetic field directly, which may improve their work efficiency and enhances the practical applicability.

Point 13: Suggest possible future research avenues building upon your work.

Response 13: Thank you for your advice. We have supplied the possible future research based on this work as follows:

(Line 291-Line 292): “How to improve the absorption capacity of biochar microspheres for oil still needs further research.”

Point 14: End your conclusion with a final statement that reiterates the significance of your study and its potential contributions to the field of wastewater treatment.

Response 14: Thank you for your advice. We have gave the potential contributions to the field of wastewater treatment in conclusion as follows:

(Line 287-Line 289): “The high removal efficiency (COD removal rate of 86.06%) and low preparing cost of biochar microsphere (Fe3O4@L-ABM500) may promote the large-scale utilization of waste biomass, and decrease the treatment cost of oily wastewater.”

Point 15: Instead of just mentioning "chemisorption," provide a brief explanation of what this term entails in this study.

Response 15: Thank you for your advice. Chemisorption is one of the common adsorption manners beside physical absorption. We have gave the brief explanation in the revised manuscript as follows:

(Line 265-Line 267): “Hence, the chemisorption deriving from the introduced alkyl chains was the main adsorption manner of Fe3O4@L-ABM500 for oily wastewater.”

Comments on the Quality of English Language

Point 16: Conduct a thorough proofread to eliminate grammatical errors and ensure consistency in formatting and style.

Response 16: Thank you for your advice. We have checked the manuscript to eliminate grammatical errors and ensure consistency in formatting and style in the modified manuscript.

Reviewer 2 Report

The article presented interesting research work in the field of wastewater adsorption, particularly for dyeing industry effluents. Being a researcher in this area, I feel several lacking in this research article.

1. The introduction doesn't contain enough cited literature (very recent ones are recommended). Also, there are long paragraphs with just one citation at the end. You must add proper references.

2. The surface charges play a very crucial job in adsorption studies. However, I can't see any investigation about the surface charges of your biochar composite and the target compounds to be absorbed. For instance, anionic attracts cationic compounds, and this way adsorption occurs chemically. However, physical and chemical phenomena together are involved in this process. Thus, I suggest deeply investigating this process to validate your research. 

3. Figure sizes are too small to see the texts on both axis. You are suggested to increase the size and resolution of all images. 

4. Why the adsorption (removal) rates are higher at acidic pH and lower at alkaline pH? Here comes the effect of potential charges. 

5. Usually the dyeing effluents contain a lot of other chemicals, it is pertinent to mention that each compound plays a role in the adsorption process. What is the applicability of your biochar composite in real textile dyeing wastewater? There are numerous studies available but very few focused on real dyeing wastewater.  

6. On the basis of the above points, I suggest major revisions to your article. 

The English language minor corrections are needed.

Author Response

Dear editor,

We would like to express our great appreciation to you and reviewers for the comments on our manuscript (Manuscript ID: materials-2580152, Title: Preparation of biochar composite microspheres and their removal ability for oil agents in dyeing wastewater). We have modified the manuscript accordingly and made revision which marked in yellow in the paper. Attached please find the revised version, which we would like to resubmit for your kind consideration as an article in Materials.

To facilitate your review, the detailed corrections are listed in the response document point by point. Both of marked manuscript and unmarked manuscript are supplied in the manuscript document. Looking forward to hearing from you.

With best regards,

Respectfully yours,

Shimin Zhai

Key Laboratory of Advanced Textile Materials and Manufacturing Technology, Ministry of Education

Zhejiang Sci-Tech University

928 the second street, Hangzhou 310018, China

Response to Reviewer 2 Comments

Point 1: The introduction doesn't contain enough cited literature (very recent ones are recommended). Also, there are long paragraphs with just one citation at the end. You must add proper references.

Response 1: Thank you for your advice. We cited more literature in the revised manuscript as follows:

(Line 55-Line 57): “Many researches have been conducted to improve the efficiency of oily wastewater treatment. Ao et al [9] prepared amine-functionalized cotton to treat oily wastewater, and the deoiling rate reached 98.5%. ”

(Line 57-Line 59): “Gholamifard et al [10] used natural zeolite and calcined bentonite to treat oily wastewater with the maximum capacity of 3.06 mg/g and 5.37 mg/g, respectively.”

(Line 63-Line 65): “Sarup et al [13] reported a superhydrophobic-oleophilic polyurethane sponge to adsorb oil from wastewater, in which the sponge was refurbished by biochar deriving from textile sludge.”

Point 2: The surface charges play a very crucial job in adsorption studies. However, I can't see any investigation about the surface charges of your biochar composite and the target compounds to be absorbed. For instance, anionic attracts cationic compounds, and this way adsorption occurs chemically. However, physical and chemical phenomena together are involved in this process. Thus, I suggest deeply investigating this process to validate your research.

Response 2: Thank you for your advice. This study emphasized the influence of hydrophilic and hydrophobic modifications on adsorption behavior. As you pointed out, the surface charges play an important role in the adsorption process, which also confirmed by the results of adsorption with different pH in present study. We will deeply investigate this process in the further research. Thank you for your advice again.

Point 3: Figure sizes are too small to see the texts on both axis. You are suggested to increase the size and resolution of all images.

Response 3: Thank you for your advice. We have adjusted the size and resolution of all images in the revised manuscript based the journal's requirements.

Point 4: Why the adsorption (removal) rates are higher at acidic pH and lower at alkaline pH? Here comes the effect of potential charges.

Response 4: Thank you for your question. Because the carboxyl groups (-COOH) (acidic pH) can be converted into -COO- at alkaline pH, which can promote its water solubility. Hence, the affinity of Fe3O4@L-ABM500 for oil decreased. In another word, the charges of functional groups influenced the adsorption abilities of biochar.

Point 5: Usually the dyeing effluents contain a lot of other chemicals, it is pertinent to mention that each compound plays a role in the adsorption process. What is the applicability of your biochar composite in real textile dyeing wastewater? There are numerous studies available but very few focused on real dyeing wastewater.

Response 5: Thank you for your question.

Point 6: The English language minor corrections are needed.

Response 6: Thank you for your advice. We have checked the manuscript to eliminate grammatical errors and ensure consistency in formatting and style in the modified manuscript.

Round 2

Reviewer 2 Report

The authors have provided satisfactory answers to some questions, but my primary question about the potential surface charges is still unanswered. It is nevertheless mandatory to provide scientific evidence to support this claim and warrant publication in Materials. 

The authors must note that such discrepancy will not attract scientific readers and thus your article will not be cited in high numbers.

The editor/s should decide in this case to accept the article or ask for another revision. I think the next revision will be useless as obvious from the answers of the authors in the current revision. 

Minor Corrections needed

Author Response

Point 1: The unit of k2 in the pseudo-second-order model is wrong. Please correct it.

Response 1: Thank you for your reminder. We have modified the unit of k2 in the revised manuscript as follows:

(Line 124): “K1 (min-1) and K2 (g/(mg•min)) are the equation constants, respectively.”

Point 2: Please provide the pore properties of resulting biochar microsphere (BET surface area, total pore volume,average pore size).

Response 2: Thank you for your suggestion. We have provided the pore properties of Fe3O4@L-ABM500 in the supporting materials and revised manuscript as follows:

(Line 134-Line 136): “The pore property of Fe3O4@L-ABM500 was characterized by a surface area analyzer (Tristarâ…¡ 3020, Mike Murray Feldman Instrument Co., USA) using the nitrogen gas adsorption isotherms.”

(Line 272-Line 274): “Besides, the pore properties of Fe3O4@L-ABM500 were also tested. The BET surface area, pore volume and pore size of Fe3O4@L-ABM500 were 3.280 m2/g, 0.003 cm3/g and 4.139 nm, respectively (See in the supporting materials).”
